# The Beneficial Effect of Eco-Friendly Green Nanoparticles Using *Garcinia mangostana* Peel Extract against Pathogenicity of *Listeria monocytogenes* in Female BALB/c Mice

**DOI:** 10.3390/ani10040573

**Published:** 2020-03-29

**Authors:** Afrah F. Alkhuriji, Nada A. Majrashi, Suliman Alomar, Manal F. El-Khadragy, Manal A. Awad, Alaa R. Khatab, Hany M. Yehia

**Affiliations:** 1Department of Zoology, College of Science, King Saud University, Riyadh 11495, Saudi Arabia; aalkhuriji@ksu.edu.sa (A.F.A.); nada430001386@gmail.com (N.A.M.); dr.alaa-rasmi@hotmail.com (A.R.K.); manalelkhadragy@yahoo.com (M.F.E.-K.); 2Doping Research Chair, Department of Zoology, College of Science, King Saud University, Riyadh 11495, Saudi Arabia; syalomar@ksu.edu.sa; 3Zoology Department, Faculty of Science, Helwan University, Cairo 11790, Egypt; 4King Abdullah Institute for Nanotechnology, King Saud University, Riyadh 11451, Saudi Arabia; mawad@ksu.edu.sa; 5Department of Food Science and Nutrition, College of Food and Agriculture Science, King Saud University, Riyadh 11451, Saudi Arabia; 6Department of Food Science and Nutrition, Faculty of Home Economics, Helwan University, Cairo 11221, Egypt

**Keywords:** *Garcinia mangostana* peel, *Listeria monocytogenes*, green nanoparticles, mice

## Abstract

**Simple Summary:**

*Listeria monocytogenes* is a resourceful foodborne pathogen triggering listeriosis. It was recently found that the bacterium caused acute and self-limiting delirious gastroenteritis in healthy individuals. In the intestinal tract, *L. monocytogenes* enters the mucosa directly via enterocytes. Animal models for *L. monocytogenes* infection have provided many insights into the mechanisms of pathogenesis, and the development of new model systems has allowed the investigation of factors that influence adaptation to the gastrointestinal environment. This study investigated that the oral administration of natural products in association with silver nanoparticles could inhibit oral infection of listeria in BALB/c mice. The effect of natural products biosynthesized silver nanoparticles (Ag-NPs) using *Garcinia mangostana* peel extract were well-known as the major targets of oral listeria infection in mice. Anti-inflammatory factors are important to treat the dangers of chronic inflammation associated with chronic diseases. The pro-inflammatory cytokine levels, such as interleukin (IL)-1 beta and tumor necrosis factor (TNF)-alpha, decreased in the intestine of mice receiving Ag-NPs using *Garcinia mangostana* peel extract. Therefore, the current study addresses the beneficial effect of the inflammation and apoptotic signaling pathways in intestine induced by *L. monocytogenes*.

**Abstract:**

*Listeria monocytogenes* is a psychrophilic bacterium, which causes widespread zoonosis in the natural environment, and mainly affects goat, sheep, and cattle herds. Recently, we predicted that it can be transmitted through food. It causes listeriosis, a severe infectious disease, which occurs with food contaminated with the pathogenic bacterium. Anti-inflammatory factors are important to treat the dangers of chronic inflammation associated with chronic diseases. Natural foodstuffs have made and are continuing to make vital contributions to the search for new antilisterial agents. The use of natural products in association with silver nanoparticles has drawn attention because of its easy, nonpathogenic, eco-friendly, and economical protocol. Hence, we aimed to biosynthesize silver nanoparticles (Ag-NPs) using *Garcinia mangostana* peel extract, which was found to be a good source for the synthesis of silver nanoparticles, their formation being confirmed by color change and stability in solution, and investigated the antilisterial activity of these nanoparticles in a murine model of *L. monocytogenes* infection. A total of 28 mice were divided into four groups—healthy control, infected, infected mice treated with green Ag-NPs biosynthesized with *G. mangostana* (5 mg/mL), and infected mice pretreated with Ag-NPs. From our results, oral treatment with Ag-NPs biosynthesized with *G. mangostana* peel extract resulted in a significant reduction in malondialdehyde (MDA), enhanced antioxidant enzyme activities, and increased the levels of the antiapoptotic protein, compared with the untreated mice. These results indicate that *G. mangostana* may provide therapeutic value against *L. monocytogenes*-induced oxidative stress and histopathological alterations, and that these effects may be related to antiapoptotic and antioxidant activities.

## 1. Introduction

The digestive tract is not only important for nutrient digestion and absorption, but it is also the largest immunological organ in the body, protecting against exogenous pathogens [1]. The endogenous gastrointestinal microbial flora plays a fundamentally important role in health and disease; critical functions of the commensal flora include protection against epithelial cell injury, regulation of host fat storage, and stimulation of intestinal angiogenesis [2]. The small intestine is able to absorb far in excess of the body’s requirements and as such, large portions of this organ can be removed without deleterious effects. However, changes in absorption and secretion homeostasis can rapidly lead to diarrhea, dehydration, electrolyte disturbance, and malnutrition [3].

The small intestine is a major site for *L. monocytogenes* invasion. *L. monocytogenes* is a foodborne pathogen responsible for a disease called listeriosis, which is potentially lethal in immunocompromised individuals. Listeriolysin O (LLO) is a toxin produced by *Listeria monocytogenes*, and it is the major virulence factor of *Listeria*. This toxin causes membrane damage. This bacterium was first used as a model to study cell-mediated immunity, and has emerged over the past 20 years as a paradigm in infection biology, cell biology, and fundamental microbiology. The *Listeria* genus includes Gram-positive, nonsporulating, rod-shaped, aerobic or facultative anaerobic microorganisms. The genus consists of more than 10 species including, *L. monocytogenes, L. innocua, L. ivanovii, L. seeligeri, L. grayi,* and *L. welshimeri* [4,5].

The main sources of listeriosis are contaminated foods such as milk and milk products. Several internal or external pathological factors, such as viral, bacterial, and parasitic infections, disrupt the oxidant/antioxidant balance, leading to oxidative stress, including the oxidation of lipids, proteins, and nucleic acids [6,7]. Accumulation of reactive oxygen species (ROS) in cells can damage membrane lipids, which are probably the most susceptible cell components, if not prevented by an appropriate antioxidant scavenging system. Several studies indicate that bacterial infections with high tolerance of the host are the result of defense mechanisms that include enhanced generation of ROS [7].

Treatment of *Listeria monocytogenes* is often difficult due to the toxic side effects, cost, and the increase in drug resistance due to extensive use of antibiotics. This antibiotic-resistant bacterium has been reported with increasing resistance over the past several decades. The antibiotic resistance of *Listeria monocytogenes* has necessitated the continued effort to identify new improved antilisterial drugs [6].

Natural products have made and continue to make important contributions to the search for new antibacterial drugs [8]. Mangosteen (*Garcinia mangostana*) is one of the most desirable tropical fruits of Southeast Asia [9]. The mangosteen plant is mainly cultivated in Indonesia, Malaysia, the Philippines, and Thailand [10], and contains secondary metabolites such as flavonoids and polyphenols. Mangosteen has recently been reported to be a rich source of a class of polyphenols known as xanthones [11]. It has many properties such as antibacterial, antifungal, antioxidant, anti-inflammatory, and antitumor activities, and shows promise for cosmetic, medicinal, oral, and pharmacological applications [12].

The use of plants as the production assembly of silver nanoparticles (Ag-NPs) has drawn attention because of its simple, rapid, eco-friendly, nonpathogenic, cost-effective protocol. They also provide large quantities of product and a single-step technique for biosynthetic processes. The reduction and stabilization of silver ions by combining them with biomolecules, such as proteins, amino acids, enzymes, polysaccharides, alkaloids, tannins, phenolics, saponins, and vitamins, which already exist in the plant extracts, have medicinal value [13]. As there is increasing interest in using nanotechnology, some reports suggest that nanoparticles (NPs), for biomedical purposes, could form the bulk of future treatment strategies for different diseases [14]. Furthermore, Allahverdiyev et al.2011 [15] found that Ag-NPs have anti-inflammatory effects by inhibiting metabolic activity through impairing mitochondrial function via oxidative stress. Therefore, in the present study, we investigated the protective role of green nanoparticles using the rind of *G. mangostana* fruit, against *L. monocytogenes*-induced oxidative stress in female BALB/c mice.

## 2. Materials and Methods

### 2.1. Plant Material and Extract Preparation

Three kilograms of fresh *G. mangostana* fruit was collected from the hypermarket at Riyadh city, Saudi Arabia, and carefully washed with deionized water several times to remove dust particles. The fruit was then separated into two parts: one part was air dried to remove the residual moisture, cut into small pieces, and stored in air-tight container. The second part was cut into smaller pieces, loaded onto a tray, and freeze dried on a shelf in a freeze dryer (Labconco 8811 Prospect Ave, Kansas City, MO 64132, USA). This was then ground into fine powder for further extractions. Freeze dried mangosteen was used in this study according to parameters determined in Appendix A.

### 2.2. Total Phenolic Content

Total phenolic compound content of *G. mangostana* extract was assayed by the Folin–Ciocalteu method as described previously [16]. Briefly, 0.1 mL of the sample’s extract was mixed with 2.5 mL of distilled water in a test tube, and then 0.1 mL of undiluted Folin–Ciocalteu reagent (Sigma-Aldrich, St. Louis, MO, USA) was added. The solution was mixed well and then allowed to stand for 6 min before adding 0.5 mL of 20% sodium carbonate solution. The color was developed for 30 min at room temperature (20 °C), and the absorbance was measured at 760 nm using a spectrophotometer (PD 303 UV spectrophotometer, Apel Co., Limited, Saitama, Japan). A blank sample was prepared using 0.1 mL of methanol instead of the extract. The measurement was compared to a calibration curve of gallic acid solution and expressed as milligram (mg) equivalent (eq.) of gallic acid per gram (g) of dry weight extract.

### 2.3. Total Flavonoids

The aluminum chloride colorimetric method was used to determine the total flavonoid content of *G. mangostana* extract as described previously [17]. Briefly, in a test tube, 50 µL of the extract was mixed with 4 mL of distilled water, 0.3 mL of 5% NaNO_2_ solution, and 0.3 mL of 10% AlCl_3_.6H_2_O. The mixture was allowed to stand for 6 min, and then, 2 mL of 1 mol/L NaOH solution was added; distilled water was subsequently added to bring the final volume to 10 mL. The mixture was allowed to stand for another 15 min, and the absorbance was measured at 510 nm. The total flavonoid content was calculated from a calibration curve, and the result was expressed as mg eq. rutin per g dry weight.

### 2.4. DPPH (2,2–diphenyl–1–picrylhydrazyl) Radical Scavenging Activity

The power of the *G. mangostana* extract to scavenge DPPH radicals was assayed as described previously [18]. A fresh solution of 0.08 mM DPPH radical in methanol was prepared. Next, 950 µL of DPPH solution was mixed with 50 µL extract and incubated for 5 min. Exactly 5 min later, the absorbance of the mixture was measured at 515 nm (PD 303 UV spectrophotometer, Apel Co., Limited, Saitama, Japan). Antioxidant activity (AA) is expressed as percentage inhibition of DPPH radical using the equation: AA = 100 − [100 × (A_sample_/A_control_)], where A_sample_ is the absorbance of the sample at time, t = 5 min and A_control_ is the absorbance of the control.

### 2.5. ABTS [2,4,6–tri(2–pyridyl)–s–triazine] Radical Scavenging Activity

The ABTS assay was used to determine the DPPH radical scavenging activity according to the method of Gouveia and Castilho (2011) [19]. The ABTS^+^ radical solution was prepared by reacting 50 mL of 2 mM ABTS solution with 200 µL of 70 mM potassium persulfate solution. This mixture was stored in the dark for 16 h at room temperature, and it was stable in this form for 2 days. For each analysis, the ABTS^+^ solution was diluted with pH 7.4 phosphate buffered saline (PBS) solution to an initial absorbance of 0.700 ± 0.021 at 734 nm.

This solution was freshly prepared for each set of analysis. To determine the antiradical scavenging activity, an aliquot of 100 µL methanolic solution was mixed with 1.8 mL of ABTS^+^ solution, and the decrease in absorbance at 734 nm (PD 303 UV spectrophotometer, Apel Co., Limited, Saitama, Japan) was recorded during 6 min. The results are expressed as µmol Trolox equivalent per g of dried extract (µmol eq. Trolox/g), based on the Trolox calibration curve.

### 2.6. Ferric Reducing Antioxidant Power (FRAP)

Ferric reducing antioxidant power (FRAP) was performed as described previously [20]. The FRAP reagent included 300 mM acetate buffer, pH 3.6, 10 mM 2,4,6–Tris(2–pyridyl)–s–triazine (TPTZ) in 40 mM HCl, and 20 mM FeCl_3_ in the ratio of 10:1:1 (v/v/v). A volume of 3 mL of the FRAP reagent was mixed with 100 mL of moringa extract in a test tube and incubated with shacking at 37 °C for 30 min in a water bath. Reduction of ferric–TPTZ to the ferrous complex formed an intense blue color, which was measured with a UV–visible spectrophotometer (PD 303 UV spectrophotometer, Apel Co., Limited, Saitama, Japan) at 593 nm after 4 min. The results are expressed in terms of mol eq. Trolox per g of dried sample (µmol eq. Trolox/g).

### 2.7. Synthesis of Ag-NPs using the Peel of G. Mangostana

Green Ag-NPs were synthesized by bioreduction of Ag+ using fresh suspension of *G. mangostana* fruit. A volume of 5 mL of the extract was added drop-by-drop to an aqueous solution of AgNO_3_ (50 mL, 0.1 mM/mL), and was stirred at 45–50 °C for 30 min. Ultrasonication was applied to the mixed solution for 3 h. The color of the silver nitrate solution changed from colorless to deep brown, indicating the formation of the Ag-NPs. The residual AgNO_3_ was removed by dialysis against deionized water at 4 °C. The Ag-NPs formed was analyzed by Zetasizer (ZEN 3600, Malvern, UK) and characterized using transmission electron microscopy (TEM) (JEM-1011, JEOL, Akishima, Japan). Furthermore, the green Ag-NP synthesis was confirmed by a UV–Visible spectrophotometer in the range of 200–1000 nm wavelength. The absorption spectra were recorded with Perkin–Elmer Lambda 40 B double-beam spectrophotometer using 1 cm matched quartz cells. Particles were prepared during 24 h and used immediately within 1 week. The stability of the Ag-NPs was examined by observing the color of the solution after 20, 40, 50, and 60 days of storage in a refrigerator at 4 °C. Mice were fed orally with 5 mg/mL of the green Ag-NPs synthesized from the rind of *G. mangostana* fruit.

### 2.8. Listeria monocytogenes Preparation and Culture

*L. monocytogenes* serotype 4a strains acquired from American Type Culture Collection (ATCC 19114), (10^10^ CFU/mL) were used to study mouse oral infection. Activated culture of *L. monocytogenes* ATCC 19114 was grown on Brain Heart Infusion Agar (Oxoid, CM 1136) and incubated at 37 °C for 24 h. One colony was picked and inoculated in Brain Heart Infusion broth (Oxoid, CM 1135), for 7 h on rotary shaker (120 RPM) at 37 °C. Optical density (O.D) at 620 nm was determined every 1 h, and at the same time, the total count was enumerated on Brain Heart Agar. After determining the cells of 10^10^ CFU/mL, (Appendix A), the cells were centrifuged (5000 rpm for 10 min); the supernatant was discarded, and the pellet was washed twice in PBS (Dulbecco A, Oxoid BR0014G, pH 7.3) and then resuspended in 10 mL of PBS. Infection dose of *L. monocytogenes* contained above 10^9^ CFU, and was orally injected to each mouse according to Angelakopoulos et al., 2002; [21] and Lecuit et al., 1999 [22]. While, the dose used by Golnazarian et al., 1989 [23] ranged from 3.74 to 6.45 log10 CFU, we challenged mice orally with 10^10^ CFU/daily using gavage.

### 2.9. Experimental Protocol

For the in vivo experiment, BALB/c female mice (n = 28; 8 weeks old) were obtained from VACSERA (Giza, Egypt). The mice were challenged with *L. monocytogenes* by oral injection of 0.3 mL of RPMI 1640 media. The animals were housed in wire-bottomed cages under standard conditions of illumination with a 12-h light-dark cycle and at a temperature of 25 ± 1 °C for 1 week until the beginning of treatment. The animals were provided with tap water and a balanced diet ad libitum. All experiments were performed in accordance with the European Community Directive (95/701/EEC). The animal care procedures agreed with the National Institutes of Health (NIH) Guidelines for the Care and Use of Laboratory Animals, eighth edition, and were approved by the Institutional Animal Ethics Committee for Laboratory Animal Care at the Zoology Department, Faculty of Science, Helwan University (Approval number: HU/Z/012-19).

The animals were randomly divided into four groups with seven animals in each group and treated as follows for 2 weeks:

**Group I:** Normal noninfected negative control group.

**Group II:** Infected untreated positive control group: mice were orally infected with *L. monocytogenes* 10^10^ CFU orally/day.

**Group III:** Infected mice with *L. monocytogenes* 10^10^ CFU orally/day and treated at the same time with green nanoparticles biosynthesized with *G. mangostana* (5 mg/mL).

**Group IV:** Mice pretreated 1 week with green nanoparticles biosynthesized with *G. mangostana* (5 mg/mL/orally/day) and then infected with *L. monocytogenes* 10^10^ CFU orally/day.

Two weeks postinfection, mice were sacrificed, and their intestinal tissues were excised promptly. Intestinal tissue samples for histopathological analysis were put in 10% formalin at −80 °C until intestinal sections were processed, and for biochemical and molecular analyses, the samples were frozen at −80 °C without formalin until processed.

### 2.10. Oxidative Stress

Homogenates of the intestine were prepared in 50 mM Tris-HCl and 300 mM sucrose to measure lipid peroxidation (LPO) in terms of the amount of malondialdehyde (MDA) formed using the thiobarbituric acid (TBA) method by Ohkawa et al. (1979) [24].

### 2.11. Enzymatic Antioxidant Status

The prepared homogenates of the intestine were used in the determination of catalase (CAT) [25].

### 2.12. Determination of Apoptotic Markers in Intestinal Tissue

Intestinal homogenates were prepared in lysis buffer and analyzed using a colorimetric caspase-3 assay kit (Product number: CASP3C; Sigma-Aldrich Co. St. Louis, MO, USA) according to the manufacturer’s instructions. The concentrations of caspase-3 in intestinal lysates were calculated with the help of the calibration curve generated using known amounts of standards. Bcl-2 (Cat. No. LS-F10920) levels were measured in the intestinal tissue lysates with ELISA kits (LifeSpan BioSciences, Inc., Seattle, WA, USA). The procedure was according to the instructions of the manufacturer. The levels are expressed as ng/mg tissue protein.

### 2.13. Real-Time PCR

Total RNA was extracted from the intestinal tissue samples using an RNeasy Plus Mini kit (Qiagen, Valencia, CA, USA). RNA was reverse transcribed using the RevertAid H Minus Reverse Transcriptase (Fermentas, Thermo Fisher Scientific Inc., Waltham, MA, USA). Real-time PCR reactions were performed using Applied Biosystems 7500 Instrument. The relative gene expression was determined with power SYBR Green (Life Technologies, Carlsbad, CA, USA) and by the comparative threshold cycle method of Pfaffl (2001) [26]. The PCR primers for *TNF*-*α* and *IL-1β* genes were synthesized by Jena Bioscience GmbH (Jena, Germany). Primers were designed using the Primer-Blast program from NCBI. mRNA levels for each sample were normalized to *GAPDH*. The following primer sets were used:
**Name****Sense (5′---3′)****Antisense (5′---3′)***TNF-α*AGAACTCAGCGAGGACACCAAGCTTGGTGGTTTGCTACGAC*IL-1β*GACTTCACCATGGAACCCGTGGAGACTGCCCATTCTCGAC*GAPDH*GCATCTTCTTGTGCAGTGCCGATGGTGATGGGTTTCCCGT

### 2.14. Histopathological Examination

Tissue samples were fixed in 10% neutral formalin for 24 h and paraffin blocks were routinely processed for light microscopy. Slices of 4–5 μm were obtained from the prepared blocks and stained with hematoxylin and eosin (H&E) as well as Masson’s trichrome for fibrosis. The preparations obtained were visualized using a Nikon microscope at a magnification of 400×.

### 2.15. Statistical Analysis

Results represent means ± standard errors of the means (SEM). Data were analyzed using the one-way analysis of variance (ANOVA). For comparison of significance between groups, Duncan’s test was used as post hoc test according to the Statistical Package for the Social Sciences (SPSS version 20.0 IBM, Armonk, NY, USA).

## 3. Results

The total phenolic and flavonoids contents of the investigated extract were found to be 11.453 ± 0.934 mg eq. gallic acid/g and 0.725 ± 0.034 mg eq. rutin/g, respectively (Table 1). Furthermore, the results revealed that the extract has potent free radical scavenging power. For the DPPH, ABTS, and FRAB assays, the values obtained were 38.564 ± 1.23, 5.892 ± 0.045, and 0.245 ± 0.0045 µmol eq. Trolox/g, respectively.

The UV-visible spectrophotometer was used in order to confirm the presence of Ag-NPs (Figure 1). As shown in Figure 1A, the graph shows the average particle size of Ag-NPs using the dynamic light scattering Zetasizer. This was the technique used to determine the size distribution profile of *G. mangostana* with Ag-NPs at 133.8 nm (Figure 1B). This result proved the homogenous distribution and size variation with no agglomeration of resulting nanoparticles, which was clearly indicated by the appearance of one peak. Furthermore, TEM image (Figure 2) demonstrated that most of the Ag-NPs were spherical or polygonal in morphology, with size up to 93.50 nm. The color of the Ag-NPs in aqueous solution did not change, indicating the stability of the Ag-NPs.

*L. monocytogenes* infection significantly (*p* < 0.05) enhanced the formation of lipid peroxidation, which implicated in the pathogenesis of intestinal tissue injury by the free radical derivatives of *L. monocytogenes* and was responsible for cell membrane damage and consequent release of marker enzymes of intestinal toxicity. Malondialdehyde (MDA), a product of lipid peroxidation that is widely used as a marker of lipid peroxidation, was significantly increased in the small intestine homogenates (Figure 3), while increase in MDA was inhibited in the green NPs-treated groups compared to the control group.

*L. monocytogenes*-infected mice (Figure 4) showed significant decrease in intestinal CAT, while it was significantly increased in green nanoparticles-treated groups, compared to the control group.

To investigate whether the observed antilisterial effects of green Ag-NPs were related to the antiapoptotic activity of green Ag-NPs, the protein levels of Bcl-2 in intestinal tissue were measured. The findings revealed that the antiapoptotic protein Bcl-2 was significantly reduced (*p* < 0.05). However, mice treated with green Ag-NPs, concurrently or prior to *L. monocytogenes* infection, showed significant increase in Bcl-2 (Figure 5).

During inflammation, the nucleus has a seminal role in immunity due to its active pro-inflammatory genes that encodes *TNF-α* and *IL-1β*. The results showed a significant upregulation in the expression of *TNF-α* mRNA and *IL-1β* mRNA in the *L. monocytogenes*-infected groups (Figure 6) compared to the control group, while the green nanoparticles-treated groups showed a significant downregulation in gene expression level.

Histological patterns observed in the small intestine (Figure 7): normal structure was observed in the control noninfected group. The infected positive group showed destructed villi, degeneration of the lamina propria, and the nuclei of the columnar cells were lost and fused together to form membranous-like shape, while the infected group treated with green Ag-NPs using *G. mangostana* showed healthy intestinal sections with well-defined and enhanced villi. Moreover, the green Ag-NPs using *G. mangostana* protected group showed degeneration in the lamina propria.

## 4. Discussion

The small intestine is an important site of infection for many enteric bacterial pathogens, and serves as the site of colonization and attachment and the seat of pathogenesis for a number of important enteric bacterial pathogens of humans and animals [27]. The use of plants as the production assembly of Ag-NPs has drawn considerable attention because of its rapid, eco-friendly, nonpathogenic, and economical protocol and they provide a single-step technique for the biosynthetic processes. The reduction and stabilization of silver ions by combining it with biomolecules, such as proteins, amino acids, enzymes, polysaccharides, alkaloids, tannins, phenolics, saponins, and vitamins, which already exist in the plant extracts, have medicinal value [28].

In this study, green nanoparticles were evaluated against inflammatory listeriosis. The results showed that green nanoparticles were able to enhance immunogenicity as demonstrated by increased survival time, reduced pathological changes in intestinal tissue, and enhanced immune response [12].

Mangosteen contains the secondary metabolites xanthones, which are polyphenols [11]. The results indicated intestinal damage, induced by *L. monocytogenes*, with the evaluation of lipid peroxidation, which showed significantly elevated levels of MDA, the end product of lipid peroxidation in the infected untreated group. Several studies have suggested that the levels of MDA increase during bacterial infections due to the membrane damage caused by cytosolic LLO [25,26,27]. In this study, the groups treated with green Ag-NPs using *G. mangostana* showed decreased MDA levels due to the action of α-mangostin from *G. mangostana* rind, which had shown to protect the mitochondria from peroxidative damages; our findings were, thus, in accordance with the results of previous studies [29,30,31].

The activities of the antioxidant enzyme CAT in mice infected with *L. monocytogenes* also decreased, wherein CAT detoxifies hydrogen to water. Thus, treatment with green Ag-NPs synthesized with *G. mangostana* extract may protect cells from damage, demise, and dysfunction caused by *L. monocytogenes*-induced oxidative stress. Our results obtained for antioxidant enzyme activities are similar to the results of other authors [13,32].

Apoptosis is initiated by various types of stimuli, including infections. To investigate the degree of apoptosis, the Bcl-2 antiapoptotic protein was measured in intestinal tissue homogenates. LLO is a pore-forming molecule, and a major virulence factor of *Listeria*. *L. monocytogenes* which induces murine lymphocyte apoptosis. The insertion of LLO into the mitochondrial membrane causes the release of cytochrome c. The insertion of LLO into the mitochondrial and/or endoplasmic reticulum membrane stimulates calcium efflux, thereby activating the calpain and/or caspases. LLO is responsible for mitochondrial network disruption along with a decrease in mitochondrial membrane potential and intracellular ATP levels. Once released, cytochrome c can bind to apoptotic protease-activating factor–1(Apaf–1) in the cytoplasm, forming a complex that can activate caspase-9 with subsequent death induction [33]. Our results showed that the levels of Bcl-2 significantly decreased in the infected untreated group, while the groups treated with green Ag-NPs synthesized with rind of *G. mangostana* showed increased Bcl-2 levels. α-mangostin contains the C10 hydrophobic tail extension, which confers potency to the compound, and is also implicated in the antiapoptotic action. Other studies have reported similar results [34,35,36].

The results of histopathological investigation showed that morphological changes in the infected group with *L. monocytogenes* at the dose of 10^10^ CFU/day for 1 week induced apoptosis and severe damage and degeneration in intestinal cells, which were in agreement with several authors (Longhi et al., 2005 [37]; Hausmann, 2010 [38]). After exposure to *L. monocytogenes*, the apoptosis can be induced by mucosal factors secreted by epithelial cells. The pro-inflammatory cytokine (TNF-α) is central to epithelial injury [38]. *L. monocytogenes* specifically targets its receptor on intestinal villi and crosses the intestinal barrier, and leads to the onset of a systemic infection [39]. This breakdown of the epithelial barrier plays a part in the disruption of epithelial defenses and further accelerates mucosal inflammation [38]). The study showed that the infected group and the group treated at the same time with green Ag-NPs using *G. mangostana* (G III) showed the best results compared with the protected group (G IV); this is due to the antioxidant and phenolic contents, particularly, α-mangostin, found in the rind of *G. mangostana* fruit that stabilize and maintain the integrity of the intestinal tissue, i.e., protect against the apoptosis damage. Our findings were similar with Chen et al., 2018 [40] and Jindarat, 2014 [41]; they found that mangosteen inhibited the pro-inflammatory cytokines and decreased TNF-α production.

The histopathological investigation revealed that *L. monocytogenes* caused progressive alterations in the small intestine; this result was in agreement with several studies (Becattini et al., 2017 [7]; Markus & Martin, 2010 [42]; Sukhadeo & Trinad, 2009 [43].

There are many cytokines participated in inflammatory response such as tumor necrosis factor alpha (TNF-α) and interleukin-1 beta (IL-1β) which both play central roles in inflammation. TNF-alpha signals through two distinct membrane-bound receptors (TNFR1 and TNFR2); the two receptors share many intracellular signaling intermediates, including nuclear factor kappa-light-chain-enhancer of activated B cells (NF-κB), and thus, together may enhance inflammatory mediator production. Furthermore, IL-1β activates similar signaling pathways as TNF-α to induce primarily pro-inflammatory gene expression (PMC2718541) [44,45]. Inflammation plays an important role in host defense, which encompasses multiple processes against external stimuli such as infection by pathogen, exposure to bacterial endotoxin, or chemical exposure.

In this study, we evaluated the effect of green Ag-NPs using *G. mangostana,* as anti-inflammatory response inhibited by signaling cascade of pro-inflammatory gene expression, which in contrast showed a significant upregulated response in the *TNF-α* and *IL-1β* gene expression levels caused by *L. monocytogenes* infection compared to the control group; this result was similar with [5,45]. Several studies have suggested that the phytochemical in green nanoparticles caused a significant decrease in the gene expression levels due to the α-mangostin found in the *G. mangostana* fruit, which is able to lower the levels of inflammatory proteins [46,47,48,49].

## 5. Conclusions

In conclusion, treatment with green Ag-NPs was more effective in inhibiting the development of intestinal toxicity by decreasing oxidative stress, downregulating pro-inflammatory cytokines genes, as well as improving the histological features of the small intestine.

## Figures and Tables

**Figure 1 animals-10-00573-f001:**
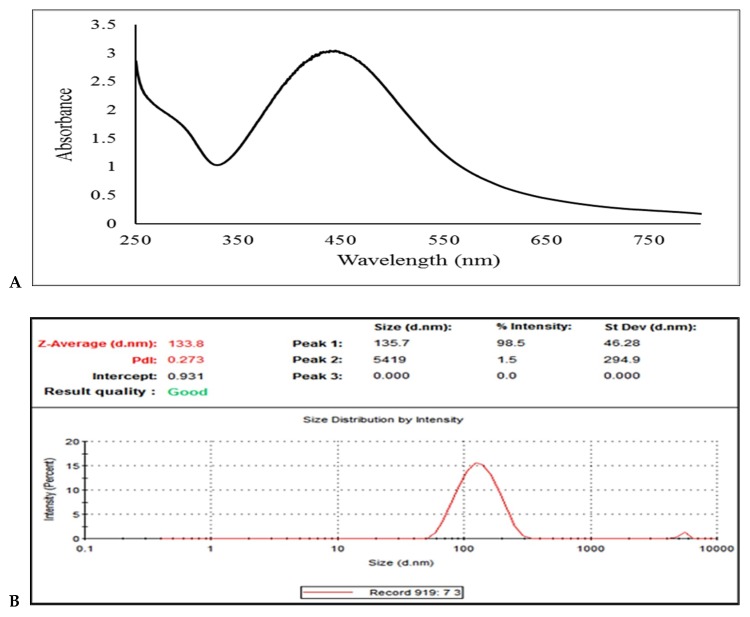
(**A**) The absorption spectrum of the green silver nanoparticles synthesized with *G. mangostana* and (**B**) a graph of a Zetasizer measurement of the average size of green Ag-NPs.

**Figure 2 animals-10-00573-f002:**
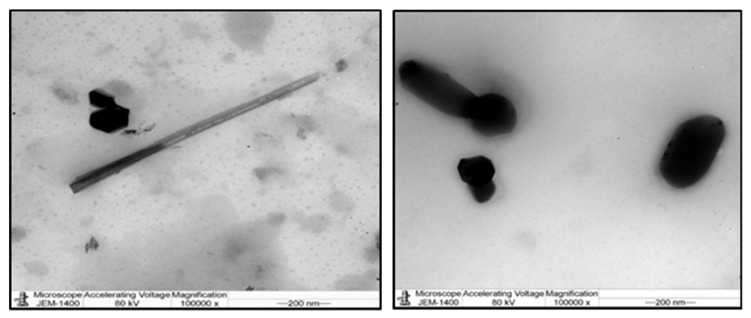
A graph of transition electron microscopy (TEM) image of green silver nanoparticles (Ag-NPs) synthesized (scale bar: 200 nm).

**Figure 3 animals-10-00573-f003:**
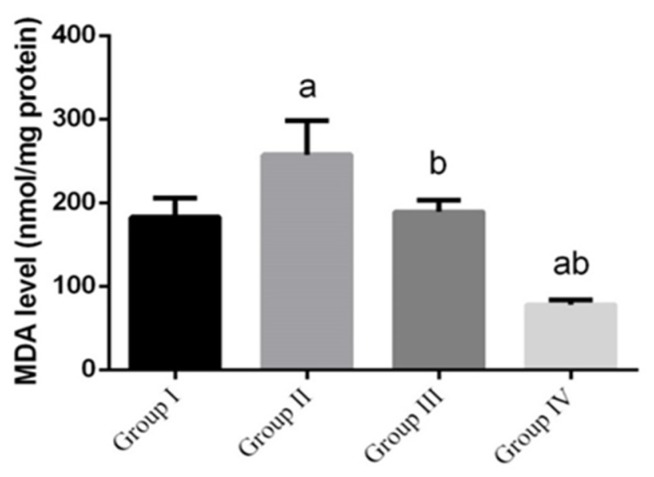
The effect of green Ag-NPs with *G. mangostana* on intestinal lipid peroxidation, (malondialdehyde, MDA) induced by *L. monocytogenes* (10^10^ CFU/day). Values are means ± SEM (n = 7). ᵃ *p* < 0.05, significant change compared to control (noninfected negative control) group (G I) and ᵇ *p* < 0.05, significant change compared to infected group (G II).

**Figure 4 animals-10-00573-f004:**
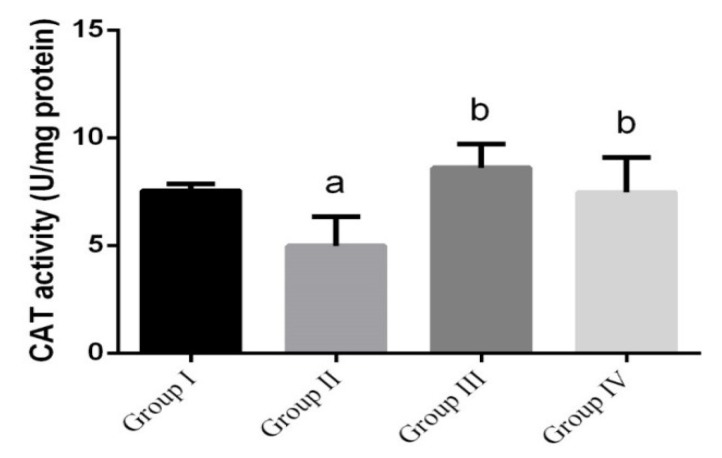
The effect of green Ag-NPs using *G. mangostana* on intestinal catalase (CAT) enzyme. Values are means ± SEM (n = 7). ᵃ *p* < 0.05, significant change compared to control (noninfected negative control) group (G I) and ᵇ *p* < 0.05, significant change compared to infected group (G II).

**Figure 5 animals-10-00573-f005:**
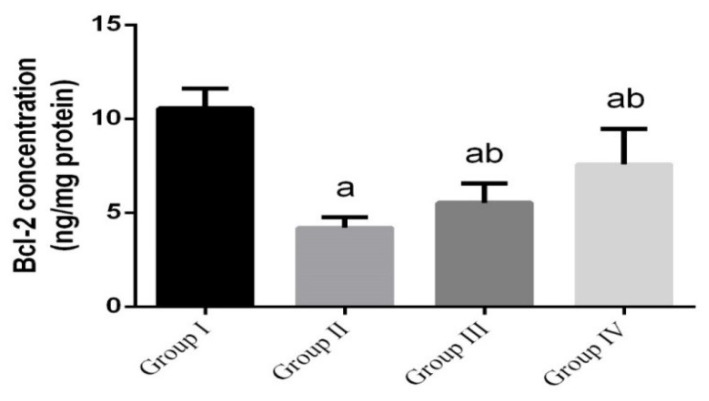
The effect of green Ag-NPs using *G. mangostana* on intestinal antiapoptotic protein (Bcl-2). Values are means ± SEM (n = 7). ᵃ *p* < 0.05, significant change compared to control (noninfected negative control) group (G I) and ᵇ *p* < 0.05, significant change compared to infected group (G II).

**Figure 6 animals-10-00573-f006:**
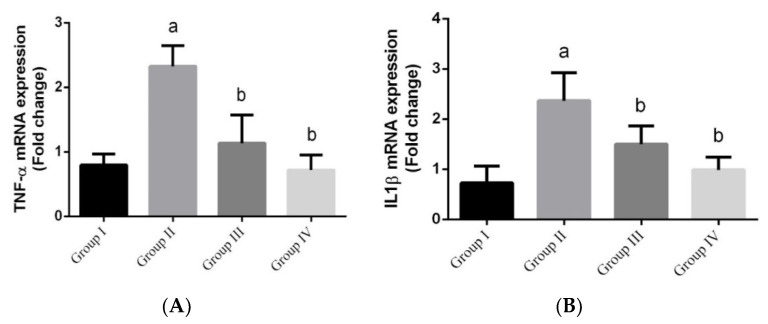
(**A** and **B**) The effect of green Ag-NPs with *G. mangostana* on intestinal gene expression of *IL-1β* mRNA and *TNF-α* mRNA induced by *L. monocytogenes*. Values are means ± SEM (n = 7). ᵃ *p* < 0.05, significant change compared to control (noninfected negative control) group (G I) and ᵇ *p* < 0.05, significant change compared to infected group (G II).

**Figure 7 animals-10-00573-f007:**
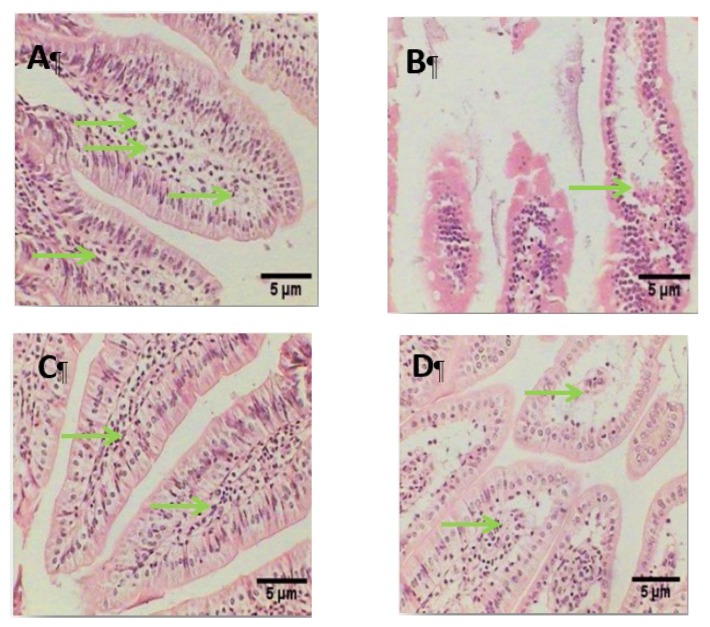
Photomicrograph of intestinal tissue stained with H&E stain (100×) showing (**A**) control group (G I), healthy structure in the mucosa layer (M), submucosa (S), muscularis, (MS), and normal healthy structure in the lamina propria (Lp) (**B**) Infected group (G II), showing a destructed villi (V) and degeneration of the lamina propria. (**C**) Infected group and treated at the same time with green Ag-NPs using *G. mangostana* (G III) showing an enhanced villus with very little degeneration of lamina propria. (**D**) The protected group, given green Ag-NPs using *G. mangostana* and after 1 week infected with *L. monocytogenes* (G IV) showing a destructed villus. Green arrows indicate lamina propria (Lp).

**Table 1 animals-10-00573-t001:** Experimental determinations of total phenolic and flavonoids contents and antioxidant capacity assays (ABTS, DPPH, and FRAB) for *G. mangostana* extract.

Parameters	Mean ± SD
Total phenols (mg eq. gallic acid/g sample)	11.453 ± 0.934
Total flavonoids (mg eq. rutin/g sample) DPPH (%)	0.725 ± 0.034
DPPH (%)	38.564 ± 1.23
ABTS (_mol eq. Trolox/g sample)	5.892 ± 0.045
FRAB (_mol eq. Trolox/g sample)	0.245 ± 0.0045

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
