# Peer review of "The Beneficial Effect of Eco-Friendly Green Nanoparticles Using Garcinia mangostana Peel Extract against Pathogenicity of Listeria monocytogenes in Female BALB/c Mice"

_animals, 2020, doi:10.3390/ani10040573_

Round 1
Reviewer 1 Report
The manuscript examines the effects of nanoparticles with Garcinia mangostana extract on Listeria induce intestinal injury (pathology). The manuscript demonstrates several interesting observations, but in this reviewer’s opinion there are sections within the manuscript that requires calcification.
Comments:
- Section 2.1: There were 2 methods (air dried and freeze dried) for G. mangostana fruit preparation. Why not only use as a single method?
- Line 170. Why was only color change an indication of nanoparticle stability. Were instrumental measurement employed to ensure product stability prior to administration to the mice. How soon following particle preparation were the particle gavaged to the mice.
- Line 199: How was the concentration of Listeria (1010 CFU/day) determined for the study. The amount of Listeria used in challenged studies can be variable- this concentration of bacteria needs to be justified
- Line 200: Was the Listeria and nanoparticles administered in the same syringe for gavage. If so could this procedure effect the viability of the Listeria bacteria. Were testes completed to ensure the bacteria remained viable. If so, supporting data (bacterial growth curves) as supplemental figures should be provided. If not then – this factor must be strongly justified
- Line 232: Only two pro-inflammatory cytokines were chosen to examine intestinal investigation. Studies in gut inflammation routinely examine many more cytokine profiles- Th1, TH2, Th17, Treg. This is a major weakness in the study, as anti-inflammatory events may be occurring (ie nanoparticle -extract treatment). The investigators must strongly justify why only 2 pro-inflammatory were measured in the experiment.
- Figure Legend 3- 6- define treatments (ie Group 1= non-infected negative control)
- Figure 6- The histology images must be redone. There is shadowing of images at the bottom of each image panel. More importantly, items discussed are not represented within the images. As an example Line 315 describe lamina propria degeneration- yet the histology images do not show the lamina propria- the images only show villi. As well, Figure 6B is not very convincing of villus degeneration- few nuclei are necrotic and it appears that there is only the occasional single cell necrosis (at least at the resolution of the images provided within my version of the manuscript).
Author Response
Dear. Ms. Crystal Zhang
Assistant Editor
Animals Editorial Office
We very much appreciated the comments and suggestions made by the reviewers. We complied with these comments in our revised manuscript.
Reviewer 1:
- Section 2.1: There were 2 methods (air dried and freeze dried) for G. mangostana fruit preparation. Why not only use as a single method?
- AU: we were compared the using extraction of air dried and freeze dried of mangosteen peels to show the antioxidants activity before doing the nanoparticles and were found that freeze dried is the most highest level of antioxidants than air dried, so the application of nanoparticles carried out on freeze dried [Table 1 in supplementary materials showed it].
- Line 170. Why was only color change an indication of nanoparticle stability. Were instrumental measurement employed to ensure product stability prior to administration to the mice. How soon following particle preparation were the particle gavaged to the mice.
Thanks for the comment. We used UV-Vis spectrophotometer in the range of 250-1000 nm wavelength to confirm the NPs synthesis. We mention in details in paragraph from line 267-270.
- Line 199: How was the concentration of Listeria (1010CFU/day) determined for the study. The amount of Listeria used in challenged studies can be variable- this concentration of bacteria needs to be justified.
AU: we determined the total viable count by growing Listeria momocytoegens ATCC19114 on brain heart infusion broth for about 7 hours at 37 °C at 120 rpm and the Optical density were measured at 620 nm every one hour and at the same time the total viable count CFU/ ml were enumerated on Brain heart infusion agar. We found that after about 6 hours the total viable count were 1010 CFU/ml. We prepared it freshly and daily to prevent the changes of viability during storage in refrigerator , beside the time for reaching this number is not more than 6 hours (Figure 1 and 2 in supplementary material showed the numbers determination) .
- Line 200: Was the Listeria and nanoparticles administered in the same syringe for gavage. If so could this procedure effect the viability of the Listeria bacteria. Were testes completed to ensure the bacteria remained viable. If so, supporting data (bacterial growth curves) as supplemental figures should be provided. If not then – this factor must be strongly justified
AU: No the manngosteeen nanoparticles were adminsterated firstly by one syringe and secondly challenged by Listeria monocytogenes ATCC 19114, by using other syringe and this in details were rewritten in the dose of each group. The bacteria prepared daily and the total count determined in the bacterial growth curve (Figure 1 and figure 2 in the supplementary material showed it.
- Line 232: Only two pro-inflammatory cytokines were chosen to examine intestinal investigation. Studies in gut inflammation routinely examine many more cytokine profiles- Th1, TH2, Th17, Treg. This is a major weakness in the study, as anti-inflammatory events may be occurring (ie nanoparticle -extract treatment). The investigators must strongly justify why only 2 pro-inflammatory were measured in the experiment.
We agree with the reviewer, there are many cytokines participated in inflammatory response. However, we just measured TNF-alpha and IL-1beta which both play central roles in inflammation as we mentioned in the discussion. TNF-alpha signaling through two distinct membrane-bound receptors (TNFR1 and TNFR2), the two receptors share many intracellular signaling intermediates, including NF-κB, and thus together may enhance inflammatory mediator production. Furthermore, IL-1β activates similar signaling pathways as TNF-α to induce primarily proinflammatory gene expression (PMC2718541). Most importantly, clinical studies have revealed that anti-TNF-α and anti- IL-1β strategies can be successful in treating inflammatory diseases. Hence, we measured these cytokines.
- Figure Legend 3- 6- define treatments (ie Group 1= non-infected negative control)
Done
- Figure 6- The histology images must be redone. There is shadowing of images at the bottom of each image panel. More importantly, items discussed are not represented within the images. As an example Line 315 describe lamina propria degeneration- yet the histology images do not show the lamina propria- the images only show villi. As well, Figure 6B is not very convincing of villus degeneration- few nuclei are necrotic and it appears that there is only the occasional single cell necrosis (at least at the resolution of the images provided within my version of the manuscript).
We mention in details in paragraph from line 323-329.
Once again we thank the reviewers for their comments and suggestions, which helped us to improve our manuscript. We hope that you can accept our revised paper as it stands now.
With best regards,
Prof. Dr. Manal Elkhadragy

Reviewer 2 Report
Aim of this manuscript is very good idea. Abstract need more results, first part is very generally. In text some words need italic type. Introduction need modification, it is lot information about liseriosis, listeria etc. and only few sentence about nanopatricles. In material and methods: 2.1 Plant material need description about how many fruits was obtained before drying and after drying. Author describe that L. monocytogenes was cultured on faculty, but ATCC is form American collection and they need write this in material and methods and after this how was culture prepared for testing. Than authors rewrite in vivo experiment, but this part need also how were feed animals, not that they have just bacteria and nanoparticles. Im not sure if 7 animals is good for some conclusion. Also I miss in study some results about microbiota of gastrointestinal tract. Results need more detailed description and discsusision ned some chronology with results. I recommend major revision of manuscript.
Author Response
Dear. Ms. Crystal Zhang
Assistant Editor
Animals Editorial Office
We very much appreciated the comments and suggestions made by the reviewers. We complied with these comments in our revised manuscript.
Reviewer 2:
- Abstract need more results, first part is very generally.
We mention in details in paragraph from line 44-48.
- In text some words need italic type.
Done
- Introduction need modification, it is lot information about liseriosis, listeria etc. and only few sentence about nanoparticles.
We mention in details in paragraph from line 98-108.
- In material and methods: 2.1 Plant material need description about how many fruits was obtained before drying and after drying.
Done
- Author describe that L. monocytogenes was cultured on faculty, but ATCC is form American collection and they need write this in material and methods and after this how was culture prepared for testing.
Yes Listeria monocytogenes ATTCC 19114 and the inoculum was prepared in our lab. (Food microbiology lab., Faculty of Home Economics, Helwan University). Also the preparation rewritten in paragraph 2.8. Listeria monocytogenes Preparation and Culture in LINES 183-194.
- Than authors rewrite in vivo experiment, but this part need also how were feed animals, not that they have just bacteria and nanoparticles.
We mention the details about the feeding of the mice in details in paragraph from line 196-199.
- Im not sure if 7 animals are good for some conclusion.
Most of authors used the same number of the mice in all PAPERS.
- Also I miss in study some results about microbiota of gastrointestinal tract.
Our target here in our paper to study the effect of mangosteen peel nanoparticles on mouse challenged by specific genus, L. monocytognes, so we focused on it, your suggestion is right and microbiota necessary changed by these nanoparticles antioxidants activity but really we did not put in our mind and need more and big study.
- Results need more detailed description and discsusision ned some chronology with results.
We mention in details in paragraph from line 378-395.
Once again we thank the reviewers for their comments and suggestions, which helped us to improve our manuscript. We hope that you can accept our revised paper as it stands now.
With best regards,
Prof. Dr. Manal Elkhadragy

Round 2
Reviewer 1 Report
Questions have not been addressed by the authors, as such these questions have been reworded for clarification. In addition and importantly, the responses to these questions needs to be added to the manuscript.
Line 170; How long following particle preparation were the particles used in the experiment. For instance, were the particles used immediately, 24 hours, 7, 14, or 21 days after preparations
Line 199: Why was the concentration of Listeria (1010 CFU/day) used in the experiment. Is there scientific evidence to support a challenge with this amount of Listeria. As an example, should the amount of Listeria been higher 1011 CFU/day or lower 109 CFU/day
Line 232: See the original question. The authors provided a response, but this justification needs to be added to the discussion. It would help explain why only a small number of specific cytokines were measures, instead of the more typical -larger cytokine panels.
Figure 7: The histological photos in my opinion still fails to illustrate the described intestinal changes. A lower magnification would allow a better appreciation of the entire intestinal morphological structure; lamina propria, muscularis, serosa. A panel of lower magnification histopathological images could be added as a supplemental figure. In addition, I believe there are newly added ‘green arrows’ to images (identifying cells) that have not been described in the figure legend; this should be adjusted.
Author Response
Reviewer 1:
- Line 170; How long following particle preparation were the particles used in the experiment. For instance, were the particles used immediately, 24 hours, 7, 14, or 21 days after preparations.
AU: The Particles were prepared during 24 hours and used immediately within 1 week, we mention in details from line 187-179.
- Line 199: Why was the concentration of Listeria (1010CFU/day) used in the experiment. Is there scientific evidence to support a challenge with this amount of Listeria. As an example, should the amount of Listeria been higher 1011 CFU/day or lower 109 CFU/day.
AU: line 194-198, According to many researchers used same microorganisms the oral injection the concentration always above 10 9 and less than 1011, so we used 1010
We added these references in the text and in references lists [21,22, 23].
- Line 232: See the original question. The authors provided a response, but this justification needs to be added to the discussion. It would help explain why only a small number of specific cytokines were measures, instead of the more typical -larger cytokine panels.
AU: Done, we mention in details from line 402-410.
- Figure 7: The histological photos in my opinion still fails to illustrate the described intestinal changes. A lower magnification would allow a better appreciation of the entire intestinal morphological structure; lamina propria, muscularis, serosa. A panel of lower magnification histopathological images could be added as a supplemental figure. In addition, I believe there are newly added ‘green arrows’ to images (identifying cells) that have not been described in the figure legend; this should be adjusted.
- Done, we mention in details from line 329-335
Once again we thank the reviewers for their comments and suggestions, which helped us to improve our manuscript. We hope that you can accept our revised paper as it stands now.
With best regards,
Prof. Dr. Manal Elkhadragy

Reviewer 2 Report
Line 183 please reject faculty it is not neccerry and it is not very right that Faculty of Home Economics
ATCC is not mean where was culture prepared but from which collection
Line 185 L. monocytogenes must be with italics and same in references.
Line 191 Oxoid
Figure 1A,B. is too small
Author Response
Reviewer 2:
- Line 183 please reject faculty it is not neccerry and it is not very right that Faculty of Home Economics.
AU: it was deleted
- ATCC is not mean where was culture prepared but from which collection
AU: American Type Culture Collection was added line 186
- Line 185 L. monocytogenes must be with italics and same in references.
AU: Done
- Line 191 Oxoid
AU: Change to Oxoid
- Figure 1A,B. is too small
AU: Done
Once again we thank the reviewers for their comments and suggestions, which helped us to improve our manuscript. We hope that you can accept our revised paper as it stands now.
With best regards,
Prof. Dr. Manal Elkhadragy
